# Attribute Based Interpretable Evaluation Metrics for Generative Models

## Abstract

While generative models continue to evolve, the field of evaluation metrics has largely remained stagnant. Despite the annual publication of metric papers, the majority of these metrics share a common characteristic: they measure distributional distance using pre-trained embeddings without considering the interpretability of the underlying information. This limits their usefulness and makes it difficult to gain a comprehensive understanding of the data. To address this issue, we propose using a new type of interpretable embedding. We demonstrate how we can transform deeply encoded embeddings into interpretable embeddings by measuring their correspondence with text attributes. With this new type of embedding, we introduce two novel metrics that measure and explain the diversity of the generator: the first metric compares the frequency of appearance of the training set and the attribute, and the second metric evaluates whether the relationships between attributes in the training set are preserved. By introducing these new metrics, we hope to enhance the interpretability and usefulness of evaluation metrics in the field of generative models.

## 1 Introduction

Significant advancements have been achieved in the image generation field, from the pioneering introduction of generative adversarial networks (GANs) to the more recent emergence of diffusion models (DMs). [5, 10, 27] In recent years, generated images are hardly distinguishable from real images. In this context, evaluating the generated images for a given training dataset has played a critical role in the development.

Envision an evaluation scenario where the outputs of two generative models are compared against a common training dataset. What would be the underlying factors for judging a set as superior to another set? As the goal of generative models is mimicking the real data distribution, various metrics have been designed to assess the similarity between the generated images and the training dataset, e.g., Fréchet Inception Distance (FID)[9], Precision and Recall[25][17], and Density and Coverage[22].

Most of these evaluation metrics capture the disparity between the training data distribution and the distribution of generated images by examining the differences in feature representations within the embedding space of a pre-trained network[26, 28]. FID is a widely used metric that quantifies the dissimilarity in visual features to assess the quality and diversity of the generated images. Specifically, it measures the distance between the real and fake distributions in the embedding space of Inception-V3[28].

An important question arises regarding the suitability of the embedding space employed for evaluating generated images. The embedding space of the pre-trained model may vary depending on the dataset and task it was trained on. For instance, Inception V3 was trained for image classification on

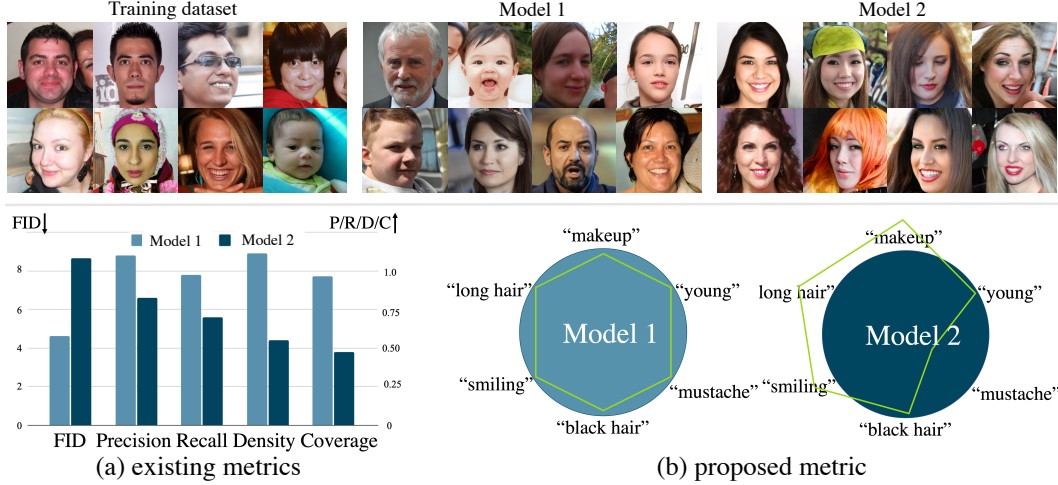

Figure 1: **Conceptual illustration of our method.** We design the scenario, Model 2 lacks diversity. (a) Although existing metrics distinguish the inferiority of Model 2, they provide no explanation about judgment. (b) Our attribute-based proposed metric has interpretation; Model 2 is biased with 'long hair' and 'makeup'.

ImageNet[3], suggesting that its embedding space is designed to compress image information and discern essential patterns for classification. Consequently, the appropriateness of employing this embedding space for evaluating generated images remains an open question.

Returning to the fundamental question at hand, Figure 1 makes an evaluation scenario a little bit more specific. Suppose there are realistic generated images from two distinct models. As shown in the example images, it is evident that Model 2 generates biased images, i.e., there are only women, while Model 1 successfully generates various images that are close to training data. Fortunately, although there remains an open question about embedding space, the values of various metrics in Figure 1 (b) align reasonably well with our interpretation; Model 1 is perceived as superior.

However, what are the underlying factors that contribute to such judgment? Although the results are consistent with a person's conclusion, it far fails to provide a comprehensive explanation. The interpretation of distances within the embedding space from a pre-trained classification model remains elusive, posing challenges in evaluation. On the contrary, humans readily discern certain factors for judgment; individuals easily recognize the bias of Model 2. These factors suggest more information and a direction beyond simple ranking. In this paper, we propose an evaluation metric that aims to interpret the underlying factors behind such judgments.

To address this objective, we begin by examining attribute comparison methods in human judgment. When evaluating two generated image distributions, humans compare the attributes present in the training dataset with those exhibited by the generated images. Key attributes under consideration include gender, facial representation, and age distribution. Ideally, with well-defined training data, we anticipate the attributes in the generated images to align with those in the training data. If the model lacks essential attributes (e.g., gender, age, glasses, or hats), it is insufficient to generate visually realistic images. Incorporating these attributes into the evaluation process may enable a more explicit and comprehensive assessment.

This paper presents a novel approach for evaluating generative models by leveraging a newly proposed embedding space that incorporates attribute-specific information. Similar to human visual judgment, our metrics evaluate images in terms of various characteristic attributes. Figure 1 (b) illustrates the concept of our metric; it captures the distribution differences of attributes. We use pre-trained CLIP [24], a language-image model trained on a huge dataset, to define a new embedding space that can quantify images for multiple attributes.

To facilitate our embedding space, we introduce the "Directional CLIPScore" (**DCS**), a method for quantifying each attribute based on the training data. Within our proposed embedding space, each channel comprises DCS values that explicitly indicate the relevance of an image to specific attributes. The use of a perceptible embedding space offers the advantage of interpretability.

We introduce two novel evaluation metrics to use the newly proposed embedding space. Firstly, the "Single attribute KL Divergence (**SaKLD**)" compares attribute distributions between training data and the generated images, providing a quantitative measure of the similarity between attribute distributions. It quantifies how closely the attributes of generated images align with the attribute distribution in training data. Secondly, we introduce the "Paired attribute KL divergence (**PaKLD**)" that considers correlations among multiple attributes. This metric accounts for the relationship between attributes, such as the presence of a beard in an image of a woman. PaKLD successfully evaluates the generated images while taking into consideration attribute relationships.

We validate our metrics through a series of carefully designed experiments, demonstrating their effectiveness and interpretability. By employing our metric, we conduct a comprehensive analysis of prominent generation models currently considered state-of-the-art [11, 13, 12, 14, 23]. Interestingly, our findings reveal variations in performance across different datasets. For instance, diffusion models exhibit superior performance on datasets with a large number of samples, such as FFHQ. In contrast, GANs outperform diffusion models on datasets with relatively smaller sample size, such as MetFaces.

In summary, this paper presents a novel approach for evaluating generative models using a new embedding space that incorporates attribute-specific information. Our proposed method, along with the introduced evaluation metrics, allows for a comprehensive assessment of generated images by considering attribute distributions and correlations. Our findings contribute to the research field by advancing the understanding and evaluation of generative models, offering insights into their strengths and limitations. Moreover, our work opens avenues for future research and potential improvements in the field of generative image synthesis by comprehensive evaluation metrics.

## 2 Related Work

**Fréchet Inception Distance**   Fréchet Inception Distance (FID) [9] measures the distance between the estimated Gaussian distributions of two datasets by passing them through a pre-trained Inception-v3[28] model. However, Kynkäänniemi et al. [18] revealed that when generated images are far from training data, the embeddings may incorrectly highlight irrelevant parts of images. To address this issue, the researchers proposed using the CLIP [24] image encoder instead of Inception-v3 to calculate the 2-Wasserstein distance, which provides reliable results regardless of the dataset being measured.

**Fidelity and diversity**   Sajjadi et al. [25] introduced precision and recall for evaluating generative model, and subsequent studies by Kynkäänniemi et al. [17] and Naeem et al. [22] have further refined this approach. Most of these methods use a pre-trained network to examine whether the embedding of generated images falls within the boundary of real image embedding (precision) and whether the embedding of real images falls within the boundary of generated image embedding (recall) for assessing fidelity and diversity.

**Rarity score**   Han et al. [6] proposed a metric for measuring the rarity of generated images. They quantified how rare the generated images are within a k-NN sphere to assess their rarity. The key difference between the rarity score and diversity in precision and recall is that the rarity score considers only the generated samples that fall within the manifold of real samples. In other words, it focuses on how well the generated images fit within the distribution of real images in terms of rarity, rather than capturing the overall diversity of generated samples.

However, we note that the concept of using raw embeddings from a pre-trained classifier remains consistent among all these metrics.

**A call for explainable evaluation**   Existing evaluation metrics in the field of generative models lack the ability to provide detailed insights into the diversity of generated images. As shown in Figure 1, even though metrics like FID, Precision and Recall indicate poor performance for a biased generator towards specific attributes (e.g., "makeup" and "long hair"), they do not provide an explanation for judgment factors. Therefore, researchers manually identified the underlying factors by visual inspection but it becomes increasingly challenging with larger sample sizes. To address this issue, we propose novel explainable evaluation metrics that provide in-depth analysis and insights into the diverse generation abilities of models.

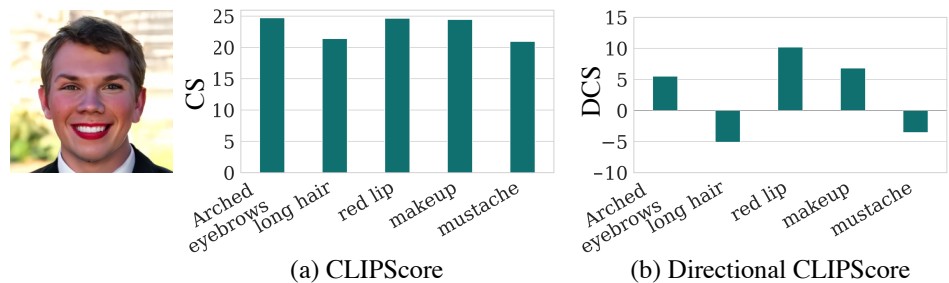

(a) CLIPScore  (b) Directional CLIPScore

Figure 2: **Difference between CS and DCS.** (a) CLIPScore[8] exhibits similar values, making it difficult to discern. (b) Directional CLIPScore has an intuitive value based on zero. We design a new embedding space; each channel represents the intensity of a specific attribute by DCS, informing explanations about the single image.

## 3 Attribute-Driven Embedding

Existing metrics for evaluating generated images commonly utilize embeddings before FCN, from Inception-V3 or CLIP image encoder[7, 4]. However, these approaches lack interpretability as the meaning of each channel in the embedding. Additionally, Kynkäänniemi et al. [18] have shown the FID scores improve significantly when the classification distribution matches that of the training set, irrespective of the quality, highlighting another limitation of the existing embedding. To address these issues and develop an explainable evaluation metric, we design each embedding of images to possess an 'interpretation'. Section 3.1 presents the process of generating explainable embedding for individual images using the CLIP encoder, and Section 3.2 introduces the Directional CLIPScore, a novel embedding approach that enhances interpretability and accuracy.

### 3.1 Attribute-driven embeddings for better representations

To achieve an interpretable embedding, we utilized each channel of the embedding as a measure of the attribute's prominence in the image. A straightforward approach to quantify attribute strength is by employing CLIPScore;

$$\text{CLIPScore}(x, a) = 100 * \text{sim}(\mathbf{E_I}(x), \mathbf{E_T}(a)),\tag{1}$$

where $x$ is a single image, $a$ is a given text of attribute, $\text{sim}(*, *)$ is cosine similarity, and $\mathbf{E_I}$ and $\mathbf{E_T}$ are CLIP image encoder and text encoder respectively. We selected multiple attributes that effectively represent image characteristics as textual descriptions and measured CLIPScore with individual images and selected attributes. The way to select attributes will refer to Section 3.3. By assigning these CLIPScores as the values for each channel in the embedding, we obtained an interpretable representation. However, relying solely on CLIPScore has challenges as the cosine similarity values tend to be similar, making it difficult to discern the relative differences between attribute strengths. Intuitively, selected human-related attributes tend to cluster closely in the CLIP embedding, resulting in smaller variations in cosine similarity. To address this limitation, subsequent subsections introduce the Directional CLIPScore, which offers a more precise scoring approach.

### 3.2 Directional CLIPScore

As discussed, CLIPScore exhibits a narrow distribution of values, which can be attributed to measuring similarity between human-related attributes, resulting in their dense clustering on the CLIP embedding. Figure 3 (a) visualizes it. To address this issue, we propose Directional CLIPScore (DCS), which leverages the centers of training images and predefined attribute texts on the CLIP embedding.

Given training data, denoted as $\{x_1, x_2, x_3, ...\} \in \mathcal{X}$, we define $C_\mathcal{X}$ as the center of images and $C_\mathcal{T}$ as another center of images for text attributes on the CLIP embedding, respectively. By using the image captioning model, BLIP[19], we define $C_\mathcal{T}$ as the center of images in text respect;

$$C_\mathcal{X} = \frac{1}{N} \sum_{i=1}^{N} \mathbf{E_I}(x_i), \quad C_\mathcal{T} = \frac{1}{N} \sum_{i=1}^{N} \mathbf{E_T}(\text{BLIP}(x_i)).\tag{2}$$

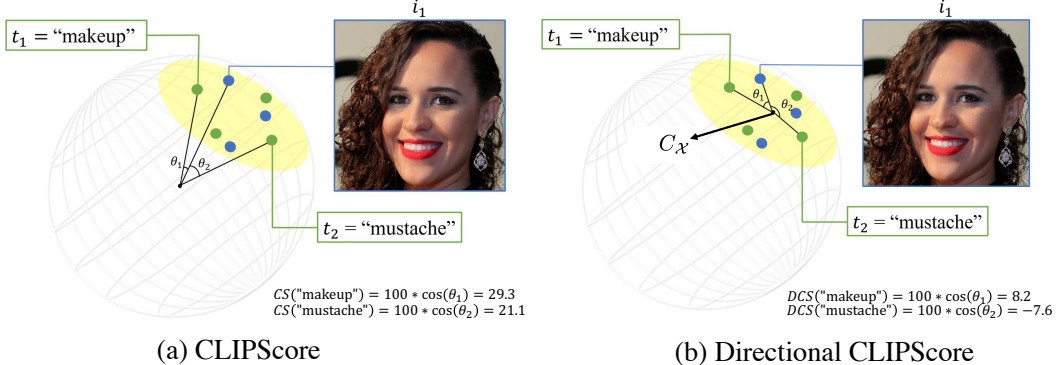

| | (a) CLIPScore | (b) Directional CLIPScore |
|---|---|---|

Figure 3: **Illustration of CLIPScore and Directional CLIPScore.** (a) CLIPScore measures the similarity between vectors with coordinate origin. (b) Directional CLIPScore measures the similarity between vectors with a defined mean of the images, $C_{\mathcal{X}}$, as the origin. In the figure, we illustrate $C_{\mathcal{X}}$ and $C_{\mathcal{T}}$ as the same point for ease of clarity and comprehension.

Table 1: **CLIPSCore and Directional CLIPScore's mean accuracy on CelebA dataset.**

| | All attributes | | Refined attributes | |
|---|---|---|---|---|
| | CLIPScore | Directional CLIPScore | CLIPScore | DirectionalCLIPScore |
| mean accuracy | 0.395 | 0.409 | 0.501 | 0.530 |

These centers serve as reference points in the embedding space and aid more accurate attribute scores. We define DCS as the measure of similarity between two directions, $V_x$ and $V_a$ where a set of attributes defined as $\{a_1, a_2, a_3, ...\} \in \mathcal{A}$. The first direction spans from the center of the image to the image itself, and the second direction extends from the center of the attributes to the desired attribute.

$$V_x = \mathbf{E_I}(x) - C_{\mathcal{X}}, \quad V_a = \mathbf{E_T}(a) - C_{\mathcal{T}}, \tag{3}$$

$$\mathrm{DCS}(x, a) = 100 * \mathrm{sim}(V_x, V_a), \tag{4}$$

where $\mathrm{sim}(*, *)$ is cosine similarity. For extending DCS from a single sample to data we denote the probability density function (PDF) of $\mathrm{DCS}(x_i, a_i)$ for all $x_i \in \mathcal{X}$ as $\mathrm{DCS}_{\mathcal{X}}(a_i)$ for brevity.

Figure 3 visually illustrates the distinction between DCS (Directional CLIPScore) and CS (CLIP-Score). Unlike CS, which lacks a clear reference point, DCS is based on the center, enabling the determination of attribute magnitudes relative to a zero point. Furthermore, DCS exhibits superior accuracy compared to CS, as demonstrated in Table 1. The table presents the accuracy results of CS and DCS for annotated attributes in CelebA[20]. By evaluating how well positive samples with the highest score align with positive samples for a given attribute, DCS consistently outperforms CS in accuracy. Notably, this trend remains consistent across refined attributes, which are removed for subjective attributes such as "Attractive" or "Blurry".

### 3.3 attribute selection methodologies

Our evaluation metric for measuring the performance of the generator is dependent on the attributes we choose to measure. To explore how to choose attributes that accurately reflect generator performance, we introduce three methods for attribute selection.

**BLIP extracted attribute**   We aim to identify and quantify the attributes present in the training data from image descriptions. We can determine which attributes are most commonly occurring in the training data by counting attributes that appear in the training data. We use the image captioning model, BLIP[19], to extract attribute-related words from training data. We use $N$ attributes that appear frequently in the training data as a set of attributes $\mathcal{A}$ for our proposed metric.

**User annotation**   Another option for attribute selection is to use a set of human-annotated attributes. By explicitly assigning attributes for evaluating generative models, users can fairly compare the

impact of each attribute score or focus on specific attributes. Especially, the CelebA dataset provides 40 binary attributes about the human face domain, which can be used to evaluate a wide range of (generated) human image sets.

**GPT attributes** We leveraged the power of GPT-3[1] to extract attributes. Through repetitive questioning, such as 'Give me 50 words of useful visual attributes for distinguishing faces in a photo' and 'Give me 50 words of useful visual attributes for discerning variations in facial features to identify people in images,' we obtained a set of attributes, which frequently appeared in the responses across different datasets. The list of questions posed to GPT-3 can be found in the Appendix, and we followed the questioning methodology outlined in [21].

# 4 Evaluation Metric with Interpretable Attribute-Driven Embedding

In this section, by leveraging the knowledge of attribute intensities, we have developed two understandable metrics. In Section 4.1, we present Single attribute KL Divergence (SaKLD), which measures the distance of attribute distributions between training data and generated images. In Section 4.2, we introduce Paired attribute KL divergence (PaKLD), a metric that assesses the relationship of attributes.

## 4.1 Single attribute KL Divergence (SaKLD)

We design SaKLD to distinguish a good generative model which produces the same quantity of each attribute present in the training data. For example, if 50,000 training data contains 3,000 images with eyeglasses, the model should generate exactly 3,000 images with eyeglasses. Any deviation from this ideal distribution is considered undesirable. We introduce a new metric that quantifies density of each attribute in dataset by utilizing interpretable embedding. Our metric, SaKLD, quantifies the difference in density for each attribute between the training dataset ($\mathcal{X}$) and the set of generated images ($\mathcal{Y}$).

We define SaKLD as

$$\text{SaKLD}(\mathcal{X}, \mathcal{Y}) = \frac{1}{N} \sum_{i}^{N} \text{KL}(\text{DCS}_{\mathcal{X}}(a_i), \text{DCS}_{\mathcal{Y}}(a_i)), \tag{5}$$

where $i$ denotes an index for each attribute, N is the number of attributes, KL(*) is Kullback-Leibler Divergence, and note that we denote the PDF of $\text{DCS}(x_i, a_i)$ for all $x_i \in \mathcal{X}$ as $\text{DCS}_{\mathcal{X}}(a_i)$.

We compare the PDFs of Directional CLIPScore for each attribute in $\mathcal{X}$ and $\mathcal{Y}$. The DCS PDF for each attribute in $\mathcal{X}$ and $\mathcal{Y}$ represent the distribution of the amount of that attribute in the respective sets. If the distribution of the amount of a specific attribute in $\mathcal{X}$ and $\mathcal{Y}$ is similar, the DCS distribution will also be similar, and the PDFs of the two sets will be close. We used Kullback-Leibler Divergence(KLD) to compare the each Directional CLIPScore PDFs for their attribute in X and Y, to quantify the extent to which the generator has created too few or too many instances of a specific attribute. We then calculate the average KLD value between the PDFs of each attribute in $\mathcal{X}$ and $\mathcal{Y}$ to obtain the final value of SaKLD.

## 4.2 Paired attribute KL Divergence (PaKLD)

We design another metric, PaKLD for examining that generated images preserve the attribute relationships present in training data. The model should generate images that adhere to the attribute relationships observed in the training data. For instance, if all 50,000 male images in the training data wear glasses, then all generated male images should also wear glasses. To evaluate the preservation of attribute relationships, we compare the difference in the joint probability density distribution of attribute pairs between training data. Our proposed metric, Pairwise Attribute KL Divergence (PaKLD), is defined with joint probability density functions as follows:

$$\text{PaKLD}(\mathcal{X}, \mathcal{Y}) = \frac{1}{M} \sum_{(i,j)}^{M} \text{KL}(\text{DCS}_{\mathcal{X}}(a_{i,j}), \text{DCS}_{\mathcal{Y}}(a_{i,j})), \tag{6}$$

where $M = nP_2$, $(i, j)$ denotes an index pair of attributes, and the pair of attributes' joint PDF is denoted as $\text{DCS}_{\mathcal{X}}(a_{i,j})$.

Table 2: **Validation of metrics by including correlated images.** The first row shows metric scores between two distinct subsets of the FFHQ dataset (30,000 images each). The rest rows show the correlated-sample-injected-scores where only one of the subsets contains an additional 300 or 600 edited images. We examine the metric performance on ("man"-"makeup") and ("man"-"bangs") correlated images. All results are average values for five random subset pairs.

| include edited images to one subset | SaKLD↓ | | | PaKLD↓ | | | FID↓ | FID$_{\mathrm{CLIP}}$↓ |
|---|---|---|---|---|---|---|---|---|
| | BLIP | USER | GPT | BLIP | USER | GPT | | |
| not included | 0.904 | 0.920 | 1.095 | 3.357 | 3.924 | 4.438 | 1.275 | 0.115 |
| ("man"-"makeup") 300 | 0.985 | 1.048 | 1.115 | 3.676 | 4.205 | 4.453 | 1.282 | 0.132 |
| ("man"-"makeup") 600 | 1.079 | 1.368 | 1.286 | 3.910 | 4.819 | 4.710 | 1.306 | 0.162 |
| ("man"-"bangs") 300 | 0.991 | 1.102 | 1.171 | 3.679 | 4.297 | 4.496 | 1.278 | 0.122 |
| ("man"-"bangs") 600 | 1.201 | 1.521 | 1.314 | 4.031 | 5.064 | 4.718 | 1.288 | 0.140 |

PaKLD analyzes the performance of the model more comprehensively. For example, if the generator's probability density function for the attribute pair ("makeup", "long hair") significantly differs from that of the training data, we can infer that the generator does not preserve the ("makeup", "long hair") relationship. PaKLD allows to quantify the degree of preservation of attribute relationships and measure quantitative entanglements between attributes that have not been considered in previous researches.

# 5 Experiments

**Experimental details**    To estimate the probability density function (PDF) of Directional CLIPScore (DCS) in the training data and generated images, we use Gaussian kernel density estimation. We sample 10,000 points from each PDF to obtain a discretized distribution and use it to calculate SaKLD and PaKLD. In all experiments, we use a set of $N = 20$ attributes.

## 5.1  Correlated Image Injection Experiment: Validating the Effectiveness of Our Metric

In this subsection, we provide a carefully designed experiment to compare the proposed metrics with FID; we first create two non-overlapping subsets of 30,000 images from FFHQ and consider them as training data $\mathcal{X}$ and generated images $\mathcal{Y}$, respectively. We then compare the scores for all metrics after including the edited images in set $\mathcal{Y}$. Specifically, we use DiffuseIT[16] to prepare two sets of edited images: 'man' with 'makeup' and 'man' with 'bangs'. We use CelebA attributes for user annotation method (denoted by USER in Table 2).

As shown in Table 2, our metrics and FID show consistent tendency: score increases when more edited images are included in imageset $\mathcal{Y}$. Furthermore, thanks to the nature of focusing on the attributes of the image domain, our metrics show more obvious numerical differences compared to FID. These results demonstrate that SaKLD successfully captures the attribute distribution difference and PaKLD captures the joint distribution difference between attribute pairs. Basically, our three attribute selection scenarios have similar tendencies across the two proposed metrics, but there are several differences. See supplement material for more details.

## 5.2  Necessity of PaKLD

We conducted another toy experiment, a scenario in which the SaKLD metric fails to detect a particular attribute relationship, while PaKLD metric successfully identified it. We define the curated subsets of CelebA-HQ as training data and generated images with discrepancies in attribute relationship. Specifically, for training data, we collect 20,000 'smiling men' images and 20,000 'non-smiling women' images using ground truth labels of CelebA-HQ. Conversely, the generated images consist of 20,000 'non-smiling men' and 20,000 'non-smiling women'. In this scenario, the PDFs of the 'man', 'woman', and 'smile' attributes would not differ significantly between the two sets, and thus the SaKLD score would not capture it well. However, Paired attribute KL divergence would exhibit significant differences because the relationships between attributes within each set are completely different.

Figure 4 clearly illustrates the disparities in the evaluation results. While SaKLD score remained relatively unchanged for noteworthy attributes such as 'man', 'woman', and 'smile', the Paired

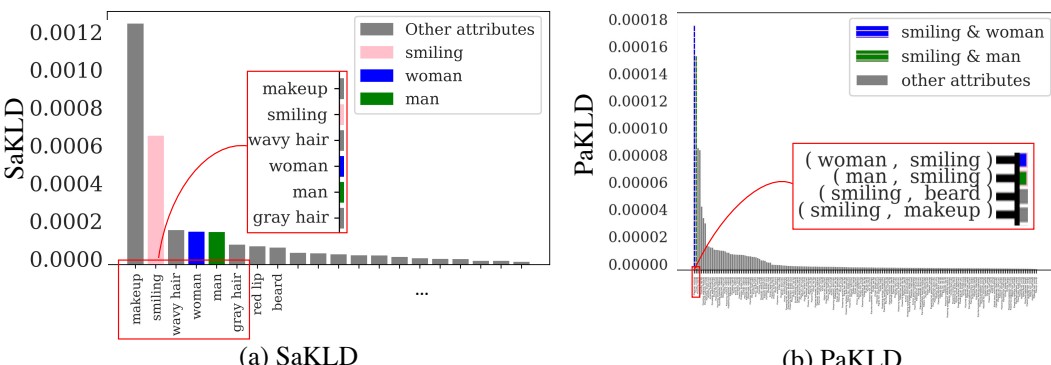

(a) SaKLD            (b) PaKLD

Figure 4: **Superiority of PaKLD.** We define the curated subsets of CelebA-HQ as training data, consisting of smiling men and non-smiling women, and generated images, consisting of non-smiling men and smiling women. (a) The most influential attribute on SaKLD is not the attribute we manipulate. (b) The most influential attributes on PaKLD provides explicit insights into the contributions of attribute pairs, such as (woman, smiling).

Table 3: **Comparing the performance of generative models.** We computed each generative model's performance on our metric with their official pretrained checkpoints. For FFHQ[11] and LSUN Cat[29], we used 50,000 images for both GT and generated set, and we used 1,336 and 50,000 images for GT and generated set for MetFaces[13]. We used BLIP-extracted attributes for this experiment.

| | SaKLD↓ | | | PaKLD↓ | | |
|---|---|---|---|---|---|---|
| | FFHQ | LSUN Cat | MetFaces | FFHQ | LSUN Cat | MetFaces |
| StyleGAN1[11] | 9.902 | 74.626 | - | 19.431 | 119.456 | - |
| StyleGAN2[13] | 6.377 | 63.601 | - | 12.838 | 100.896 | - |
| StyleGAN2-ADA[12] | 14.118 | - | 40.769 | 21.930 | - | 87.118 |
| StyleGAN3[14] | **5.993** | - | **31.140** | **12.285** | - | **58.065** |
| iDDPM [23] | - | 110.229 | - | - | 136.579 | - |
| iDDPM(P2) [2] | 12.040 | - | 129.627 | 21.507 | - | 230.720 |

attribute KL divergence score showed significant variations. This can be attributed to the distinct probability density functions (PDFs) of the 'woman ∩ smiling'. Note that we can easily understand the judgment factors; top attributes such as 'woman ∩ smiling' and 'man ∩ smiling' increase the score. These findings demonstrate the superior sensitivity and discernment of our proposed metrics, allowing for a more comprehensive evaluation of the generator's generation ability.

## 5.3 Comparing generative models including GANs and diffusion models with our methods

Leveraging the superior sensitivity and discernment of our proposed metrics, we compare the performance of GANs and Diffusion Models (DMs) in Tables 3. Interestingly, there are two attractions; 1) StyleGAN2-ADA shows the worst performance and 2) despite the respectable generative capability of DMs, iDDPM showed worse performance than StyleGAN models in all datasets.

The score of StyleGAN2-ADA implies that data augmentation for generative models may ruin attribute distribution in spite of FID's superiority. Please refer to Appendix for an analysis. And we suppose that although there are many advantages of DMs, it is inferior to GANs in attribute-based analysis.

To investigate the reason for the inferiority of DMs, we leverage the flexibility of constructing attributes to analyze the score changes according to the characteristics of attributes. We constructed attributes that focus only on color (e.g., 'yellow fur', 'black fur') and attributes that focus on shape (e.g., 'pointy ears', 'long tail') for LSUN Cat.

Table 4 shows that iDDPM's performance was particularly poor for color attributes. This is consistent with the assumption by Khrulkov et al. [15] that the encoder map of DMs coincides with the optimal transport map for common distributions; which means the pixel-based Euclidean distance corresponds to high–level texture and color–level similarity regardless of dataset and model. Therefore, the color

Table 4: **Computing performance of models with different attributes for LSUN Cat.** Analyzing the weakness of iDDPM for specific attribute types, such as color or shape. We used BLIP-extracted attributes for this experiment.

| | color attributes | | shape attrbutes | |
|---|---|---|---|---|
| | SaKLD↓ | PaKLD↓ | SaKLD↓ | PaKLD↓ |
| StyleGAN1[11] | 36.614 | 75.884 | 33.214 | 72.454 |
| StyleGAN2[13] | **36.621** | **67.518** | **34.642** | **68.954** |
| iDDPM [23] | 111.302 | 121.877 | 72.181 | 80.511 |

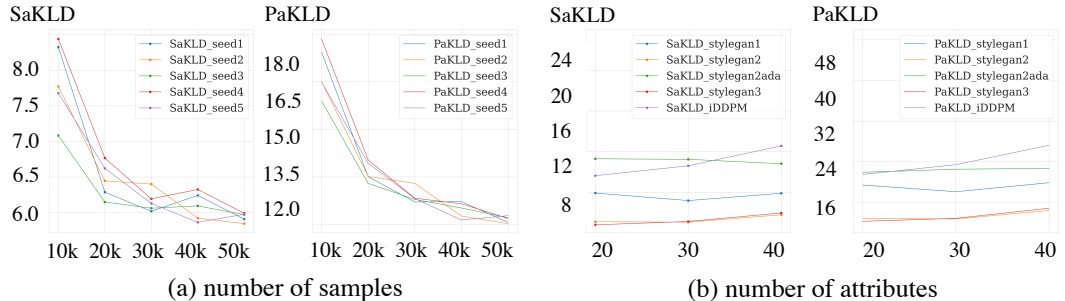

(a) number of samples        (b) number of attributes

Figure 5: **(a) The effect of sample size on our metric.** Proposed metrics started to stabilize when using more than 50,000 images. **(b) The effect of the attribute counts on our metric.** Although depending on the characteristics of the additional attributes, the ranking of scores between models can vary, the rank of the models mostly remained consistent regardless of the number of attributes.

of the output images only depends on the initial latent noise $x_T$, and the Monge optimal transport map between training data and the standard normal distribution. We conclude that the distribution of color-related attributes is the inferiority of DMs.

## 5.4 Impact of Sample Size and Attribute Count on Proposed Metric

We provide ablation experiments to investigate the effect of a number of samples and attributes in Figure 5. We obtain generated images by StyleGAN3 from FFHQ with various random seeds. When the number of samples increases, SaKLD and PaKLD converge, especially more than 50,000 samples (Figure 5 (a)). We argue that the scores started to stabilize when using more than 50,000 images and note that we use 50,000 images for Tables 3 and 4. As for the number of attributes, we observe that the rank of the models mostly remained consistent regardless of the number of attributes. However, scores of DMs, purple line of Figure 5 (b), is increased as the number of attributes is increased because of color-related attributes. We argue that 20 attributes are sufficient, but more information can be obtained by using more diverse cases. Please see Appendix for an analysis of each score.

## 6 Discussion and Conclusion

In this paper, we introduce a novel metric that not only assesses the performance of the generator but also provides explicit explanations. Our proposed method, Directional CLIPScore, quantifies the attributes captured in an image and aligns them close to human judgment. Leveraging the interpretability of DCS, we propose two novel metrics, namely the SaKLD and PaKLD, which allow us to compare attribute appearance frequencies and examine attribute relationships, respectively.

While our metrics offer comprehensive explanations, unreliable results may arise when the attributes present in the images are ambiguous. For instance, in complex modern artworks with intricate color patterns, extracting appropriate attributes becomes challenging or even impossible, rendering our metric ineffective. Additionally, if the generative model's ability is significantly poor, the same limitation arises: measuring DCS from generated images becomes challenging.

Despite these limitations, our research establishes a solid foundation for the development of explainable evaluation metrics for generative models and contributes to the advancement of the field.

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
