# OpenReview forum: "Attribute Based Interpretable Evaluation Metrics for Generative Models"
_NeurIPS.cc/2023/Conference — Submitted to NeurIPS 2023_

### Official Review · Reviewer_E51D · 2023-06-15

**Soundness:** 3 good
**Presentation:** 3 good
**Contribution:** 3 good
**Rating:** 6
**Confidence:** 4

**Summary:**

Paper proposes an evaluation metric for generative models which compare the distributions of real and generated images using a predefined set of attributes, or pairwise occurrences of attributes. The advantage of these metrics over the previous work is that they provide explicit visibility of which aspects contribute to the final metric value.

**Strengths:**

The idea for the proposed metric is novel – it uses language-image models to measure alignment of distributions of real and generated images given a set of text attributes. Additionally, the metric is highly customizable for downstream tasks because users may define their own set of attributes that are of interest to the specific task and drop irrelevant attributes.

**Weaknesses:**

Failure modes of the metric are not discussed (or limitations of language-image models and how they affect the metric).

Consider a scenario where there are two models, A and B, with the same SaKLD. Model A produces essentially perfect alignment on all attributes other than one which fails dramatically, causing a large spike in SaKLD histogram, and this attribute is the only contributing to the final score. Model B on the other hand performs poorly across all attributes but averaging over attributes yields the same SaKLD score as the model A. In this scenario SaKLD would potentially not agree with human judgment, since failing in a single attribute might not be visible when inspecting large image grids. Can this kind of scenario occur in practice, and if it can, what would be your recommendation for the user of the metric in that case?

Fig. 4 shows that SaKLD and PaKLD are dominated by few attributes of attribute pairs. Is this usually the case in practice? Fig. 5 (b) also indicates that adding new attributes contribute to the metric with diminishing strength. This might be misleading for the user of the metric. Intuitively, adding a large set of attributes should correspond to more thorough evaluation of the model, however, this might not be the case if few attributes are dominating the final value of the metric.

The empirical effectiveness of the attribute based metric is not fully demonstrated. The authors advocated for an interpretable metric but unfortunately end up comparing modern generative models using single scalar numbers (as the existing metrics do), instead of taking advantage of the interpretability of the metric and showing a more fine-grained analysis of the models.


**Questions:**

1. Fig. 2 shows CS and DCS values for distinct attributes, how would these scores look like for pairs of “opposite attributes”, e.g., “long hair - short hair” or “makeup - no makeup”? Would DCS values be symmetric around zero or something else?

2. The description of Tab. 1 was slightly unclear, how is Tab. 1 exactly calculated? The numerical values seem to be relatively close to each other, is the difference here significant? Tab. 1 shows mean accuracy but what does the accuracy distribution looks like?

3. Automatic extractions of text attributes is an interesting idea, how robust BLIP is with various datasets? What are the most frequently extracted attributes?

Minor notes:

4. Sec. 3.3 a capital letter is missing
5. Line 154: Needs to be “the center of the images”
6. Line 220: $n$ and $P_2$ are not defined
7. Tab. 3 would benefit from also having a column for FID to see how SaKLD and PaKLD differ from it


**Limitations:**

The authors adequately addressed the limitations of their work.

---

> ### Author Rebuttal · Authors · 2023-08-10
>
> ### Failure modes of metrics
>
> Here are 2 failure modes we anticipate. We add these into More Discussion.
>
> #### Bias in CLIP may deliver inaccurate results.
> As review o9jM referred, if some attributes are highly correlated in CLIP embedding space, SaKLD, and PaKLD will not resemble human judgment. Embeddings from biased encoder will be plotted as distorted distribution causing unreliable results. Using a biased model is inevitable since all models are biased including CLIP. However, we do not directly use the features but utilize them to calculate the distance between distributions. Even though the model is biased, the distance is meaningful since the features from the real dataset are also extracted by that biased model.
>
> Even though the model is biased, the distance is meaningful since the features from the real dataset are also extracted by that biased model.
>
> #### Need enough samples
> We use Gaussian Kernel Density Estimation(Gaussian KDE) to make Probability Density Function(PDF) for each attribute from the dataset's DCS. If the number of images is too small, the subset's PDF may not describe the full dataset's PDF, and it could deliver an inaccurate interpretation of the full dataset (or the generative model's performance). Fig. 5 (a) describes the impact of sample size and 50k images are recommended to get accurate results.
>
> ### Drastic failure in single attribute v.s. balanced failure among all attributes
> It is difficult to determine model A (drastic failure in a single image) or B (balanced failure among attributes) is worse, since each user has different demand. So, instead of being buried only on the total scores of SaKLD or PaKLD, we recommend inspecting SaKLD (or PaKLD) to get a thorough insight into the model. One can easily look at the top-scoring attributes from SaKLD or PaKLD. Furthermore, the main strength of our metric is flexibility, which contains users can
> drop some careless attribute (even shown significant SaKLD), or add some attribute which one mainly concentrates.
>
> ### Dominance of several attributes
> We carefully mention that we set up an extreme experiment(man-smile correlation=1) to show some dramatically poor attribute SaKLD(PaKLD) in experiment 4. Therefore, dominated by a few attributes of attribute pairs is our desired result.
> In other words, if the dominating phenomenon is observed, it can be considered to have as much impact as the extreme experiment we designed.
>
> And in practice, a similar phenomenon is observed among current major models. As we described in global rebuttal 3., each model has shown insufficient performance mimicking some attribute distributions.
>
> In the same context, we can find some of the model's total SaKLD/PaKLD drop when the number of attributes increased, as Figure 5. (b). We can understand this phenomenon as some attribute they lack is inserted.
>
>  ### Exploring major models with our metrics
> [W4] The empirical effectiveness of the attribute-based metric is not fully demonstrated. The authors advocated for an interpretable metric but unfortunately end up comparing modern generative models using single scalar numbers (as the existing metrics do), instead of taking advantage of the interpretability of the metric and showing a more fine-grained analysis of the models. -->
> We added exploration results of major models in global rebuttal 3, and also Appendix C. We reported each model's characteristics, such as iDDPM having a particularly high SaKLD for "women" and "makeup". As interpreting the model is one of our main contributions, we moved the interpretable analysis part from the Appendix to Section 5.
>
>
> ### DCS symmetric about 0?
> DCS values for opposite attributes are heading in opposite directions in common, but not the same as the absolute value. Considering DCS embedding space, an angle between embedding for "long hair" and embedding for "short hair" may not be absolute $180^\circ$ since  DCS embedding depends on text mean. We get text mean via considering not only "long hair", and "short hair", but for other attributes, the DCS for the opposite attribute cannot be symmetric as ideally. Nevertheless, signs are always observed to be opposite, which brings ease of interpretation.
>
> We attached figures of CS, and DCS for opposite attributes for 2 images in the .pdf file. Please refer to Figure 3 in the global rebuttal PDF file.
>
> ### Understanding accuracy
>
> We perform binary classification over all attributes in CelebA according to DCS and compare it with the ground truth attribute labels. Positive means having an attribute. Higher accuracy means DCS agrees with the annotation. Thank you for the constructive comment. We will clarify this as follows.
>
> Here are the first 19 attribute accuracy among 40 attributes. Attributes such as "Double_Chin"'s accuracy is <0.2 for both CS and DCS, while explicit attributes such as "blonde hair"' accuracy is almost 0.7 for DCS. We attached the whole 40 attribute's accuracy in the Appendix.
>
> ### More information about BLIP attribute extraction
> The list below is the top 20 frequent attributes in BLIP caption sorted by descending order. BLIP provides a standardized attribute extraction for all datasets but also has a shortage. As BLIP does not describe an image in great detail, for example, 'a cat sitting on a table in a room was commonly seen in LSUN cat captions, we recommend users to use USER annotation mainly.
>
>
> LSUN cat : ['a cat', 'a black cat', 'the floor', 'cats', 'a couch', 'a black and white cat', 'a white cat', 'a couple', 'a woman', 'a table', 'the ground', 'a dog', 'a man', 'black and white cat', 'a small kitten', 'a person', 'blue eyes', 'a blanket', 'a chair', 'a kitten']
>
> #### Minor notes
> Thank you for helping us improve our paper. And for $_nP_2$ in L222, it meant a permutation of two out of $n$. We have reflected them in the paper and ran another proofreading.

---

> > ### Comment · Reviewer_E51D · 2023-08-15
> >
> > Thank you for your detailed answers to my feedback. I especially appreciate addressing the failure modes and providing further empirical exploration with a larger number of generative models. At this stage, I do not have further questions for the authors.

---

### Official Review · Reviewer_ch3Y · 2023-07-04

**Soundness:** 1 poor
**Presentation:** 3 good
**Contribution:** 1 poor
**Rating:** 4
**Confidence:** 2

**Summary:**

This paper proposes a new metric to evaluate the quality and diversity of generated images based on
interpretable embeddings. To obtain the interpretable embeddings for selected attributes, the cluster
centers of the encodings of two separate encoders, one for images and the other for the text attributes
are calculated and the interpretable embeddings are the difference between the encodings of each
image and attribute and their respective cluster centers. The interpretable embeddings represent the
direction in which a particular embedding lies with respect to it’s cluster center. The CLIPScore
between the interpretable embeddings of an image and an attribute is calculated and is named as
’Directional CLIPScore’, since the interpretable embeddings represent the ’direction’.
The authors have proposed two metrics : 1. SaKLD - to quatify how closely the attribute distri-
bution in generated images matches with that of training images. 2. PaKLD - to quantify correlations
between different attributes. The KL divergence between the probability density functions of the
Directional CLIPScores for all the images in the train data and generated data.
Using SaKLD, the KL divergence between the attribute distribution in training and generated
images is calculated. PaKLD calculates the KL divergence similar to SaKLD, except that the presence
of a pair of attributes is required in the training and generated images.


**Strengths:**

The motivation for the idea is good and has a huge potential impact for improving evaluation of generative models.

**Weaknesses:**

- Although, the motivation for developing this metric is valid, the overall methodology and the
experimental results are not convincing for the use of this metric. Also, the experiments performed
are inadequate and do not sufficiently justify how well the metric performs compared to the previously
proposed metrics. Additionally, previous metrics can directly evaluate quality of generation based
on the generated images alone, but this metric heavily relies on the attributes in the form of text
descriptions. Thus, it limits the applicability and generalizability of this metric.
- The proposed metrics use CLIPScore (which already exists) for interpretable embeddings and then
applies KL Divergence for the PDFs of the ClipScores of images and attributes, thus, showing limited
novelty.
- Most of the paper is easy to follow but some important parts like, how is the center of text attributes
calculated, results from table 1 (what does accuracy stand for) etc. are a bit ambiguous. There is a
lot of scope to improve the technical soundness of the paper. Although some of the popular metrics
1are mentioned, there has been a lot of work in the generative modelling domain which the literature
survey must cover. The proposed methodology is not very sound and is not well supported with the
experimental setup. The results are also not sufficiently explained. Diversity is the main motivation
for the paper as it is mentioned in the abstract but any theoretical or empirical work to support it is
completely missing.
- Motivation and methodology including the steps to evaluate generated samples is understandable but
some parts are ambiguous (please refer above comments).
- Based on the current state of the experiments, the contributions don’t seem to be significant as there
is not enough validation to support the claims.

 typos:
- Line 43 : Instead of ”Figure 1 (b)”, it should be Figure 1 (a)
- Line 168 : Section 3.3 First letters of all the words in the heading must be capital
Other remarks :
- Line 56 : Instead of ”If the model lacks essential attributes” the following sounds technically
correct ”if model lacks ”representation” of essential attributes”.
- Line 56-58 : This claim does not seem to be correct.
- Line 147 : Link to the specified figure is missing
- Figure 2 is never refered to in the text, is it an unnecessary figure?
- Line 253 : Generated set has non-smiling men and non-smiling women. But Figure 4 caption
says otherwise

**Questions:**

1. If I understood correctly, the text and the image encoders are different? If yes, then
the embedding space of both models is going to be very different and will be influenced by the
downstream tasks that they are trained on. Would this metric always work for embeddings from
different models? Does it make sense to calculate similarity between embeddings from different
models?
2. Table 1 represents accuracy results for CS vs DCS. What is meant by evaluating ”positive
samples” in line 164 and 165? And what is meant by ”how well they are aligned”? Also, what
exactly is the accuracy calculated for?
3. In Section 3.1, there is a mention of ”each channel of the embedding being utilized”, how exactly
does this happen?
4. Line 196-197 claims that the generated distribution is desirable only if it has exactly same number
of samples as in the training set that has a particular attribute. Why should it be? Afaik, the
underlying distribution of latents represent semantic features of the data and the generative
model is responsible to generate novel samples that could be realistic but not necessarily show
exactly same features same number of times in the training set. So, is the assumption in the
mentioned lines appropriate? Can you prove this or maybe refer to some papers that provide
such guarantees?

**Limitations:**

limitations have been addressed

---

> ### Author Rebuttal · Authors · 2023-08-10
>
> # Reviewer ch3Y
>
> ### Experiments should be more convincing
> > Although, the motivation for developing this metric is valid, the overall methodology and the experimental results are not convincing for the use of this metric.
>
> We appreciate recognizing the validity of our motivation for evaluating attributes of generated images. We could not understand what is missing in our experiments from the review. If it is meant to be how our metrics vary across different image distortions, we do not focus on image quality and diversity because they can be measured by existing metrics. In order to strengthen our ground, we add one more experiment with omitting eyeglasses where SaKLD decreases according to number of omitted images. Could you elaborate if there are more to add?
>
>
> |                                                                                   | SaKLD | PaKLD  | most influencing attribute for SaKLD |
> |-----------------------------------------------------------------------------------|-------|--------|--------------------------------------|
> | eyeglasses 3325/total 50000  vs eyeglasses 3260/ total 50000 | 0.632 |  3.421 |                "beard"               |
> |  eyeglasses 3325/total 50000  vs eyeglasses 2000/total 50000 | 0.892 |  4.050 |             "eyeglasses"             |
> | eyeglasses 3325/total 50000  vs  eyeglasses 1000/total 50000 | 1.545 |  5.668 |             "eyeglasses"             |
> |  eyeglasses 3325/total 50000 vs eyeglasses 0/total 50000  | 3.257 | 11.595 |             "eyeglasses"             |
>
>
> ### Comparative experiments to previous metrics
> We first remind that the main advantage of our metrics is the interpretability where users can observe which attributes are properly modeled or messing up. Previous metrics cannot measure this aspect. On the other hand, our metrics measure how well per-/pair-attribute distribution of generated images align with the training images (Figure 4, S2, S3, S4, and S18). Furthermore, Table 2 compares how FID and FID$_\text{CLIP}$ changes over different number of injected images. We will add precision and recall in the table during the discussion period.
>
> ### Relying on textual attributes
> We would love representing attributes in the image modality but it is prohibitive because an image cannot hold exactly *one* attribute because an image is composition of various attributes. Textual attributes are common and plausible modality for representing attributes in images[LLaVA]
>
> [LLaVa] Liu et al. Visual Instruction Tuning, arXiv, 2023
>
> ### CLIPScore and KLD are not novel
> We propose directional CLIP score (DCS) instead of CLIPScore. DCS is superior to CLIPScore in attribute classification as shown in Table 1. Furthermore, DCS leads to signed scalar over binary attributes as shown in Figure 2. These improvements make SaKLD and PaKLD more sensitive to attributes in the images. KLD is a common measure for computing divergence of one distribution from another. We respectfully remind that we focus our novelty on defining the distributions of attribute strengths from a set of images rather than using KLD.
>
> ### Some important parts are a bit ambiguous
> ##### Center of attributes
> Eq. (2):
> $C_{\mathcal{T}} = \frac {1}{N} \sum_{i=1}^{N} E_{\textbf{T}}(\text{BLIP}(x_i))$,
> where $E_{\textbf{T}}$ is CLIP text encoder and {$x_i$} are training images. BLIP produces a caption for a given image $x_i$. $E_{\textbf{T}}$ produces a CLIP text embedding of the caption. The rest computes their mean. We are not sure which part is ambiguous. Could you elaborate?
>
> ##### CelebA attribute classification accuracy
> We perform binary classification over all attributes in CelebA according to DCS and compare it with the groundtruth attribute labels. Positive means having an attribute. Higher accuracy means DCS agrees with the annotation. Thank you for the constructive comment. We will clarify this.
>
> ##### Method is not sound
> > There is a lot of scope to improve the technical soundness of the paper.
> > The proposed methodology is not very sound and ...
>
> We could not understand which part is not sound. Could you elaborate?
>
> ##### Missing literature
> > Although some of the popular metrics 1are mentioned, there has been a lot of work in the generative modelling domain which the literature survey must cover.
>
> We include commonly used metrics for in the related work section: FID, FID$_\text{CLIP}$, precision & recall, improved precision & recall, density & coverage, rarity score. We will add perceptual path length, Fr\'echet segmentation distance as below. Inception score is omitted because it is barely used currently. We also omit the metrics for measuring alignment between condition (input) and generation (output) such as mIoU because our goal is to measure the divergence of generated images from training images. Could you mention any other relavent works that would be helpful to include?
> >> Perceptual path length [PPL] measures sum of perceptual distances between samples along latent interpolation to indicate smoothness of the latent space. Fr\'echet segmentation distance [FSD] compute Fr\'echet distance between the number of pixels of segmented categories in fake images and real images.
>
> [PPL] Karras et al., A Style-Based Generator Architecture for Generative Adversarial Networks, CVPR 2019
> [FSD] Bau et al., Seeing What a GAN Cannot Generate, ICCV 2019
>
>
> ##### Method is not supported by experiments
> > The proposed methodology is ... not well supported with the experimental setup.
> > Based on the current state of the experiments, the contributions don’t seem to be significant as there is not enough validation to support the claims.
>
> * Superiority of DCS over CLIPScore is supported by Table 1.
> * Misaligning attributes are captured by SaKLD as shown in Figure 4a.
> * How SaKLD reflects injecting irrelevant attribute is shown in Table 2.
> * Misalinging pairs of attributes are captured by PaKLD as shown in Figure 4b.

---

> > ### Author Response · Authors · 2023-08-10
> >
> >
> > Could you elaborate which part is not supported?
> >
> > ##### The results are also not sufficiently explained
> >
> > We could not catch what results are not sufficiently explained. Could you elaborate?
> >
> > ##### Diversity is the main motivation
> > > Diversity is the main motivation for the paper as it is mentioned in the abstract but any theoretical or empirical work to support it is completely missing.
> >
> > We respectfully remind that our main motivation is interpretable evaluation rather than diversity as written in L39-L51.
> >
> > ### typos
> > Thank you for helping us improve our paper. We have reflected them in the paper and ran another proofreading.
> >
> > ### Image and text embeddings
> > > [Q1] If I understood correctly, the text and the image encoders are different? If yes, then the embedding space of both models is going to be very different and will be influenced by the downstream tasks that they are trained on. Would this metric always work for embeddings from different models? Does it make sense to calculate similarity between embeddings from different models?
> >
> > We use pretrained CLIP image and text encoders for producing image and text embeddings, respectively. CLIP encoders are trained to maximize cosine similarity between an image and its matching text while minimizing cosine similarity between non-matching pairs of image and text. Hence, they live in the same embedding space. It does not make sense to calculate similarity between embeddings from *different models*. It is why we use CLIP embeddings.
> >
> > ### Meaning of positive sample and alignment
> > It is merged in a previous item marked with [Q2].
> >
> > ### Meaning of each channel of the embedding
> > > [Q3] In Section 3.1, there is a mention of ”each channel of the embedding being utilized”, how exactly does this happen?
> >
> > The procedure for encoding an image into an interpretable embedding is as follows: 1) compute similarities between an image and $N_A$ attributes using DCS. 2) form an $N_A$-dimensional embedding with the similarities. We will add this at the very beginning of Section 3.1. Thank you for the constructive comment.
> >
> > ### The same number of samples with specific attribute?
> > > [Q4] Line 196-197 claims that the generated distribution is desirable only if it has exactly same number of samples as in the training set that has a particular attribute. Why should it be?
> >
> > Please consider an extreme example. If we have a dataset containing dogs and cats, a generative model that produces only dogs is clearly underfit because it assigns no probability to cats. [DeepLearningBook] [Page 715.](https://www.deeplearningbook.org/contents/generative_models.html)
> > We will tone down our phrase as below.
> > >> [before] We design SaKLD to distinguish a good generative model which produces the same quantity of each attribute present in the training data. For example, if 50,000 training data contains 3,000 images with eyeglasses, the model should generate exactly 3,000 images with eyeglasses. Any deviation from this ideal distribution is considered undesirable. We introduce a new metric that quantifies ...
> > >> [after] If we have a dataset containing dogs and cats, a generative model that produces only dogs is clearly underfit because it assigns no probability to cats [DeepLearningBook]. In this context, we consider a generative model better than another if it produces similar number of samples along attributes. As we do not have an access to the groundtruth real and fake distributions, we design a new alternative metric, SaKLD, that quantifies ...
> >
> >
> >
> > [DeepLearningBook] Goodfellow et al., Deep Learning, MIT Press 2016

---

> > > ### Comment · Reviewer_ch3Y · 2023-08-14
> > >
> > > Thank you for your detailed response. Some of the questions are clarified by the answers but there are still some points I would like to mention which are a bit ambiguous regarding the experiments and literature.
> > >
> > > - there is a Figure 2 in the paper that compares CLIP Score with DCS, but It is not clear whether the figure represents an illustration, in which case it would be a bit redundant because there is another illustration of the DCS score in Figure 1 or are the actual scores for the given image. In any case, I did not find any reference to this figure in the paper.
> > > -  the ablation experiment described in section 5.4 states that the scores depicted in figure 5a stabilize after 50k samples but the figure shows scores only until 50k samples. In that case, it would be better to show some more results with more number of samples to support the claim.
> > > - the literature covers different metrics used for evaluation of generated samples. Some metrics like Precision-Recall or Density-Coverage address the quality as well diversity aspects of a generative model which the proposed metric does not consider. If not, a plausible explanation would fit in well.

---

> > > > ### Author Response · Authors · 2023-08-15
> > > >
> > > > Figure 1 compares DCS of two distributions. It illustrates what we aim to measure.
> > > > On the other hand, Figure 2 compares CLIPscore and DCS of one sample. It shows that DCS is more suitable for measuring strength of attributes than CLIPscore.
> > > >
> > > > Experiments with more than 50K samples is now running. We will share the results in approximately 24 hours.
> > > >
> > > > We assume that existing metrics are currently sufficient for measuring quality, thus we focus on other aspects.
> > > > Our metrics measure how closely a fake distribution emulates the diversity of a real distribution using the KL divergence between the real and fake PDF of attribute strength. If the real distribution has wide or narrow spectrum of smile, the fake distribution should have wide or narrow spectrum of smile to achieve low SaKLD, respectively. The same principle applies to pairs of attributes in PaKLD. Furthermore, SaKLD / PaKLD provides which attribute distribution deviates from the real distribution. In contrast, Precision-Recall and Density-Coverage do not provide any clue about the type of diversity being measured.
> > > >
> > > > Thank you for the constructive discussion. We will add these in the revised version.

---

> > > > > ### Author Response · Authors · 2023-08-16
> > > > >
> > > > > We report the SaKLD and PaKLD of StyleGAN3, sampled with 5 different seeds in the table below. It confirms that SaKLD and PaKLD are stable around 50k images.
> > > > >
> > > > > While PaKLD may further decrease after 70k and higher image quantities result in greater precision, we recommend using 50k images for praticality.
> > > > >
> > > > >
> > > > >
> > > > >
> > > > > |       |           | 10k          | 20k          | 30k          | 40k          | 50k          | 60k          | 70k          |
> > > > > |-------|-----------|--------------|--------------|--------------|--------------|--------------|--------------|--------------|
> > > > > |       | seed1     | 8.27         | 7.99         | 7.83         | 7.88         | 7.77         | 7.84         | 7.80         |
> > > > > |       | seed2     | 8.34         | 8.20         | 8.09         | 8.15         | 8.10         | 8.06         | 8.03         |
> > > > > | SaKLD | seed3     | 9.13         | 8.43         | 8.04         | 8.04         | 7.88         | 7.84         | 7.80         |
> > > > > |       | seed4     | 8.66         | 8.20         | 7.89         | 7.98         | 7.85         | 7.89         | 7.87         |
> > > > > |       | seed5     | 8.40         | 7.93         | 7.61         | 7.79         | 7.66         | 7.74         | 7.68         |
> > > > > |       | mean(var) | 8.56(0.123)  | 8.15(0.039)  | 7.89(0.036)  | 7.96(0.019)  | 7.85(0.026)  | 7.87(0.013)  | 7.83(0.016)  |
> > > > > |       |           |              |              |              |              |              |              |              |
> > > > > |       | seed1     | 23.47        | 21.18        | 20.29        | 20.08        | 19.69        | 19.69        | 19.45        |
> > > > > |       | seed2     | 23.62        | 21.54        | 20.60        | 20.43        | 20.17        | 19.95        | 19.76        |
> > > > > |       | seed3     | 24.90        | 21.98        | 20.48        | 20.15        | 19.67        | 19.45        | 19.30        |
> > > > > | PaKLD | seed4     | 24.29        | 21.70        | 20.44        | 20.23        | 19.80        | 19.67        | 19.48        |
> > > > > |       | seed5     | 23.82        | 21.03        | 19.82        | 19.88        | 19.48        | 19.44        | 19.19        |
> > > > > |       | mean(var) | 24.02(0.337) | 21.48(0.148) | 20.32(0.092) | 20.15(0.040) | 19.76(0.065) | 19.64(0.043) | 19.43(0.046) |

---

### Official Review · Reviewer_o9jM · 2023-07-06

**Soundness:** 3 good
**Presentation:** 3 good
**Contribution:** 3 good
**Rating:** 6
**Confidence:** 4

**Summary:**

The embedding space used for calculating FID is computed with Inception V3 which is trained for image classification as the target task. This means it is more likely to capture discriminative features, raising doubts about its ability to effectively evaluate generative models. There is also a need to devise a new evaluation metric that can interpret underlying factors. This paper introduces a method called Directional CLIP Score (DCS) to properly evaluate this. The pre-trained CLIP is used as the embedding space. In particular, the paper proposes "Single attribute KL div (SaKLD)" to measure single attribute alignment and "Paired attribute KL div (PaKLD)" to measure multiple attribute alignment as new metrics. This paper provides some insightful measurement result using the proposed metrics.

**Strengths:**

The effort to evaluate generative models from various perspectives through the introduction of such metrics can be considered novel and a contribution. Especially considering the current issues such as the bias in stable diffusion [1], the proposal of such metrics can bring benefits to the field from the perspective of trustworthy AI.

**Weaknesses:**

It appears to be an intrinsic limitation that a significant number of samples (50k) are still required to obtain stable results. Nonetheless, thanks to the reported findings, we can gain more insights, and I appreciate that.
Remaining concerns are written in [Questions] section.

**Questions:**

1) In lines 96-97, it seems that the authors of the referenced paper already propose using the CLIP embedding space for evaluating generative model? If not, it would be helpful to clarify the differences compared to previously proposed methods. Regarding the for explainable evaluation in the related work section, it is necessary to determine whether this paper is the first to come up with it. If there are any additional previous works, it would be helpful to share the,

2) The major concern raised is the reliance on the embedding space. What if there are biases in the CLIP space itself? For example, if being female and wearing makeup are highly correlated, even a woman who does not actually wear makeup may show high similarity to makeup attribute. In such cases, it becomes difficult to consider CLIP score or CLIP direction score as accurate measures of similarity. While a more disentangled multimodal latent space may help alleviate this problem, I'm curious about the authors' perspective on this issue.

3) Directional CLIP Score adopts a method of computing the center of training images and normalizing them. Does this mean that anyone who wants to evaluate a generative model needs to share all these training images? Also, considering the introduction of the auxiliary image captioning model called BLIP, the improvement in performance seems minimal. Do the authors believe it is worth the trade-off?

4) Question about Table 5: cI may have missed it, but was there an explanation for the last column of FID_clip in Table 5? What does it represent? Also, in Table 5, it seems that SaKLD is calculated for a single attribute. How is it computed?

5) Can PaKLD be proposed for more than two attributes? It would be great if judgments could be made for a wide range of attributes as shown in Figure 1 (b).

6) How should we interpret numerical values itself of the scores?  Although it may not be as interpretable as accuracy I'm still curious about the authors' insights based on their experience.


**Limitations:**

It seems that there are no particular specific limitations.

---

> ### Author Rebuttal · Authors · 2023-08-10
>
> ### Image quantity's impact on result stability
>
> We use Gaussian Kernel Density Estimation(Gaussian KDE) to make Probability Density Function(PDF) for each attribute from dataset's DCS. If the number of images is too small, the subset's PDF may not describe the full dataset's PDF, and it could deliver inaccurate interpretation of full dataset (or generative model's performance). Fig. 5 (a) describes the impact of sample size and 50k images are recommended to get accurate result.
>
> ### First explainable metric
>
> >Regarding the for explainable evaluation ... it is necessary to determine whether this paper is the first to come up with it.
>
> While our evaluation protocol and $FID_{CLIP}$ both use CLIP embedding, they are different as follows. FID$_\text{CLIP}$ directly computes FID on CLIP embedding space. The CLIP embedding is still an uninterpretable feature vector of 512 dimensions, and each channel of the embedding does not have its explicit meaning. On the other hand, we compute similarity between an image and a set of given attributes in CLIP embedding space. It projects the uninterpretable CLIP embedding of the image to an interpretable embedding space, and each channel of our final embedding conveys similarity of the image to its respective attribute.
> This paper is the first to come up with an explainable evaluation which is applicable to various settings. We will add a closely related work [GANseeing] as follows.
>
> [GANseeing] Bau et al., Seeing What a GAN Cannot Generate, ICCV, 2019
>
> ### What if CLIP is biased?
>
> Including CLIP model, every model may be biased. However we utilize biased pretrained models as a feature extractor for evaluation metric because we usually calculate the distance of distribution of the features. Even though the model is biased, the distance is meaningful since the features from real dataset are also extracted by that biased model.
> Nevertheless, it is ideal that there is no bias as much as possible. We suppose that if there are biases in the CLIP space itself, CS and DCS both are not accurate but DCS is better than CS since we move the origin into the middle of attributes.
>
>
> ### Do we need all training images?
>
> Obviously, it is recommended to use all the training images as the same as in other evaluation metrics. For efficiency, as FID does, one can store and reuse CLIP embeddeings of all images. Also, one can use the well-designed subset of training dataset or evaluation dataset.
>
> ### Is the captioning model worth the trade-off?
>
> One of the advantages of our metric is its flexibility; one can proceed with the desired task and analysis. Using the auxiliary image captioning model is a smart approach that removes costly manual annotation. We kindly mention that improving the performance is not the purpose of using the captioning model. We believe it is worthwhile not only for the efforts to annotate but also for a standardized evaluation way.
>
> ### Explanation of FID$_\text{CLIP}$
> FID is Frechet distance between real embeddings and generated embeddings on Inception-V3’s penultimate feature space. $FID_\text{CLIP}$ computes the same Frechet distance but on CLIP image embedding feature space. We included FID and FID$_\text{CLIP}$ in Table 2 to show that they somewhat correlate (negatively) with the number of injected images. Still, they do not provide any clue for what attributes are under-/over-represented. While the original CLIP embedding space is not interpreted, our proposed attribute embedding is designed for interpretable by exploiting the superior text-image model.
>
> ### More than 2 attributes with PaKLD
> Of course, our metric can calculate more than two attributes. We conduct the Triple-attribute KLD between GT FFHQ images and generated images from iDDPM. And we observed the probability P("a person" & "glasses" & "a cell phone") was the most significant difference in 3D joint probability between GT and generated images. We made judgments using diverse combinations of attributes and added it into Appendix. However, we kindly mention that intuitive interpretation is difficult in more than three attributes.
>
> ### Interpreting numerical values of metric
>
> We can interpret numerical values by looking in the top-scoring attributes. We briefly share our insights in Appendix. C, and also decide to move some information into main paper. Please refer to global rebuttal.
> In Appendix.C, we computed major model’s score, and StyleGAN3 was ranked as the best model among all when using BLIP extracted text attributes. Additionally, while comparing each generative model’s score is helpful, a more thorough interpretation of each model’s performance via SaKLD or PaKLD would be beneficial. For example, StyleGAN3 failed to capture training images “eyeglass” or “hat” distribution, possibly due to its training approach of alias-free modeling between fine attributes.

---

> > ### Comment · Reviewer_o9jM · 2023-08-15
> >
> > Thank you for the detailed responses. They helped clarifying my understanding a lot.
> >
> > About the response to "What if CLIP is biased?", author's response makes sense to some extent. I still think that if CLIP is biased, it will encode image of a "man" and word "mustache" closely which may make difficult to tell if the high similarity is because the generated image truly include "mustache" or not. I guess this question cannot be perfectly resolved since the latent space of CLIP is quite unknown. I'd like to note that this did not affect my rating.
> >
> > As of now, I have no further questions and I keep the rating.

---

> > > ### Author Response · Authors · 2023-08-16
> > >
> > > We agree that bias in CLIP may lead to inaccurate CS or DCS results, leading to biased interpretation. We will discuss its impact within the discussion section of the paper.
> > >
> > > While an unbiased encoder would be the ideal solution, addressing this issue is a complex and extensive task to form another solid research topic of fairness. It will be an interesting avenue to check if techniques from fairness literature may improve robustness of our metrics.
> > >
> > > Thank you for engaging in healthy discussion.

---

### Official Review · Reviewer_1cWg · 2023-07-07

**Soundness:** 3 good
**Presentation:** 3 good
**Contribution:** 2 fair
**Rating:** 5
**Confidence:** 3

**Summary:**

This is a very interesting research work. The main contribution of this paper is to consider the attribute information in the original training data when evaluating the quality of images generated by the model. There are two benefits to this approach: 1) determining whether the model can correctly imitate the distribution of the training data; 2) explaining which attributes the model does not perform well on. In implementing this idea, the authors found that directly calculating the CLIPScore between the image representation and the text representation of the attribute does not yield distinctive results. Therefore, they proposed Directional CLIPScore (DSC). The main idea of this approach is to move the reference point for calculating vector similarity to a more reasonable point. At the same time, they proposed two methods to apply DSC, one is Single attribute KL Divergence, and the other is Paired attribute KL Divergence (PaKLD) considering the combination of attributes.

**Strengths:**

1. I think this research question is very interesting. It is very meaningful to use the attribute information of the dataset to help evaluate the quality of the model's generation.
2. The Directional CLIPScore (DSC) proposed by the authors is very concise and appears to be quite effective in the case study.

**Weaknesses:**

1. In Section 3.3, I agree with the extraction of attributes using the BLIP and manual annotation methods. However, one method of extracting attributes is to generate an attribute list with GPT and then filter it with training data. I think the attribute list generated by GPT in advance may bring biases. The core of this paper is mainly to study the correlation between the model and the training data. However, if the list generated by GPT is not the most representative attribute, the results may be biased.
2. The experimental part in Sections 5.1-5.3 does not seem very convincing. The authors mainly verify that the proposed method can indeed be consistent with some expected experimental designs, but there is a lack of more convincing quantitative indicators to show that their proposed evaluation metrics are better than those proposed by others previous research works. I would prefer to see the authors analyze the correlation coefficient between their proposed evaluation metrics and human evaluation, as well as whether their evaluation metrics have improved in terms of correlation coefficient compared to previous evaluation indicators. This is my biggest concern for this work.


**Questions:**

1. In Section 3.1, can the problem exhibited by CLIPScore be solved by directly calculating the Euclidean distance between E(x) and E(a) without relying on Directional CLIPScore?
2. Notation overload. The meanings of N used in Eq. (5) and Eq. (2) are different.

**Limitations:**

see weakness

---

> ### Author Rebuttal · Authors · 2023-08-10
>
>
> ### GPT may bring bias
> We tone down our claim regarding the use of GPT in the paper as follows.
> Instead of being a vital network for obtaining text attributes, GPT is positioned as a recommendation module to assist users in selecting attributes. As users cannot manually apprehend all the visual attributes in training datasets, external models including GPT can be helpful but should be treated as a tool, rather than the auxiliary modules in the attribute selection process.
>
> ### Correlation with human judgement
> We respectfully note that collecting human evaluation for generated attributes is impractical due to the number of samples and subjectiveness. While image quality and diversity can be roughly evaluated by humans with maybe 100 images, attribute distribution evaluated with 100 images is prone to hasty generalization. Furthermore, evaluating 50K images for attributes is impractical. As a remedy, we offer a novel evaluation protocol for broad and customizable attributes using interpretable embedding and divergence of a generated distribution from the real distribution.
>
> ### Euclidean CLIPScore
>
> >can the problem exhibited by CLIPScore be solved by directly calculating the Euclidean distance between E(x) and E(a)?
> >
> No, it cannot be resolved by Euclidean distance because CLIP is trained with cosine distance. To confirm it, we provide additional experiments in the table below. As expected, Euclidean cases are inferior to cosine cases in CelebA attribute classification.
>
>
> |   | Euclidean | Cosine |
> | -------- | -------- | -------- |
> | As-is embedding     | 0.222     | 0.395     |
> |Directional embedding| 0.225 | 0.409 |
>
> #### Notation error
> >Notation overload. The meanings of N used in Eq. (5) and Eq. (2) are different.
>
> We replaced N used in Eq. (5) to another notation to avoid overload in Eq. (2). Thanks for finding.

---

> > ### Author Response · Authors · 2023-08-19
> > **We need your comment**
> >
> > We appreciate with your thoughtful comments. Could you check our response? We will be happy to answer follow-up questions if any.

---

### Official Review · Reviewer_T7Dd · 2023-07-25

**Soundness:** 2 fair
**Presentation:** 2 fair
**Contribution:** 2 fair
**Rating:** 6
**Confidence:** 3

**Summary:**

This paper proposes two new metrics allowing to measure and explain the diversity of a generated set of images w.r.t a training set. Instead of the usual distributional distances relying upon an embedding space from a pre-trained model, these metrics rely on a set of textual attributes. The similarity between an image and an attribute is computed using their representation in a common semantic space, via the CLIP model - vectors are shifted using a centre of training images/attributes to make similarity scores more meaningful. Several ways to obtain attributes (Captioning, User-based or GPT-based) are investigated. The usefulness of the metrics is tested with an experiment injecting images correlated with target attributes in data, an experiment aiming to detect a specific attribute relationship in a curated dataset, and in a comparison of several generative models.

**Strengths:**

- This paper proposes an interesting approach, aiming at using the common semantic space between images and text proposed by CLIP to measure attribute and image relatedness, in order to provide interpretable representations of generated images and measure attribute-based metrics between a reference and generated dataset; such an application seems relatively original to me.
- The two first experiments demonstrate well the usefulness of the approach, able to detect a shift when the data has been curated by human, based on which attributes.

**Weaknesses:**

- The presentation of the paper could be improved upon. This include the writing, and the readability of the figures, as well as the quantity of information provided. This last points concerns mainly the presentation and framing of the problem of interpretability of representations, as well as the presentation and motivation of the experimental settings.
- In particular, the paper lacks related work on concept-based representation for interpretability of images. While building metrics dedicated to generative model seems new to me, there are based on an idea which has been explored extensively before. See for example "Concept Whitening for Interpretable Image Recognition, Chen et al, 2020".
- The paper focuses on a narrow choice of methods to generate attributes, which, to me, should be one of the key experimental investigation of the paper. Notably, the previous literature explores using different kind of attributes, coming from existing data (for example, "Interpretable Basis Decomposition for Visual Explanation, Zhou et al, 2018") or to be learnt ("A Framework to Learn with Interpretation", Parekh et al, 2021). The authors only (very shortly) argue about the number of needed attributes.
- The toy experiments seem relevant but are very fastly presented and should be expanded upon. The remaining experiments are too short to be convincing and only focus on a handful of models.

**Questions:**

- Could you better formalize the notation $DCS_{\mathcal{X}}(a_{i,j})$ for joint distributions ?
- Is it necessary to center the text and images differently ? They lie in the same semantic space. Also, would it not be reasonable to be using the attributes themselves, rather than captions ?


**Limitations:**

- Previous distributional metrics are several times referred as relying on external models in your paper. However, the attributes that you use also rely on external models (except the USER one, of course - but in this case, the computing of DCS still relies on a captioning model). How would you address this issue ?

---

> ### Author Rebuttal · Authors · 2023-08-10
>
>
> # Reviewer T7Dd
>
> ### Presentation of paper can be improved
>
> [W1-1] We modify the writing and figures for readability. Please refer to attached PDF file for figure. As mentioned in global announce 1, we also add more quantity of information. Please refer to global rebuttal.
>
> [W1-2] Evaluating different models allows users to choose a model that meets their needs. FID, precision, recall, density, and coverage measure quality and diversity so that users can choose a model with the best quality or compromise quality with diversity. SaKLD and PaKLD measure how much the generated images align with the training images regarding semantic attributes. For that, directional CLIP score interprets an image as amounts of attributes. Hence, users can choose a model with the desired attributes.
>
> [W1-3] We provide our motivations and improve the presentation as followings:
>
> ex 5.1 Validation of our metric's effectiveness:
> To check whether our metric operates properly, we intentionally injected  "man with makeup" images only into an experimental set. We found deteriorations on  both SaKLD and PaKLD regarding "makeup" since they are rare in the training set. Total SaKLD and PaKLD scores have worsed also, which reflect both individual attributes/attribute affected to total score. Similarly, SaKLD regarding a specific attribute linearly decreases by excluding images having the attribute from one of the sets.
>
> ex 5.2 Ablational validation of PaKLD
> For thorough validation, we established a particular scenario where setA consists of images with more smiling men than unsmiling men and more unsmiling women than smiling women, and in setB, it is vice versa.
> We set both groups to have the same number of images for each attribute ("smiling", "man" and "woman"). While SaKLD scores regarding each attribute were similar, PaKLD regarding "smiling man" and "smiling woman" showed significnat differences compared to the other PaKLD scores. It demonstrates that PaKLD accurately understand the difference between the correlation of attributes. We used celebA datasets for the experiment, since it has ground truth attribute labels.
>
> ### Concept based representation
>
> Concept Whitening extracts human-understandable concepts from black-box models through the CW layer. In certain contexts, the motivation behind interpreting black box models closely aligns with our metric. However, our interpretation approach diverges significantly as it is posthoc. We recognize the significance of associating our problem set with a concept-based representation, going beyond mere posthoc methods, in order to derive more valuable insights. We organized it and put it in the related work.
>
> ### Further exploration of attribute selection
> In addition to language-text models like BLIP and GPT, the incorporation of methods such as classifier model's heatmap for obtaining attributes and ranks can introduce new attributes, potentially leading to novel perspectives in understanding generative models. We concur with the utilization of a variety of methods for attribute extraction. Beyond this, our primary contribution lies in our pioneering effort to interpret generative models. While attribute extraction methodologies may yield broader feedback, they are not as central as our primary objective: interpreting models through attribute quantification and comparative analysis. Additionally, the reason we set the number of attributes up to 40 attributes because celebA has 40 attributes. If a user is interested in hair styles, one can enrich the target attributes. We have added this topic to Discussion.
>
> ### More explanation for toy experiments
>
> We add more detailed explanation which contains the content in global rebuttal and [W1-3]. We designed the toy experiments to destroy the distribution in the way we want. The outcomes of this experiment, particularly the injection of "man&makeup" images that are rarely found in the training set, leads to a noticeable linear increase in both SaKLD and PaKLD scores. However, there were no distinctive patterns observed when we injected another ground-truth subset.
>
> ### Text mean and Image mean
>
> #### Dividing text mean and image mean
> It is necessary to center text and images differently. Appendix B.1. provides an ablation experiment: setting the text origin as the text mean vs image mean. We additionally provide the entire mean setting in the table below. Using the text mean shows the best accuracy. The image origin is set to the image mean.
>
> | | different text/image mean | same text/image mean | entire mean |
> | :--------: | :--------: | :--------: | :------: |
> | accuracy     | 0.409     | 0.228        |   0.313 |
>
> We suppose the image embedding and the attribute embedding are slightly unaligned because CLIP is trained with sentences and we use a simplified form of “a photo of {attribute}”. In other words, a single attribute does not fully describe one image. This is our motivation for using different centers for images and attributes.
>
> #### Text mean as attribute themselves
> Thank you for constructive suggestion. We verified using attribute themselves for obtaining $C_{\mathcal{T}}$ is more accurate than previous method in CelebA experiment. We conducted all experiment in our paper and confirm that there is no change in tendency. Please refer to global rebuttal. Thanks again for great suggestion.
>
> |  | captions | attribute themselves |
> | :--------: | :--------: | :--------: |
> | accuracy     | 0.409     | 0.442     |
>
>
> ### Relying on external models such as BLIP
> We respectfully suppose a misunderstanding. We do not object using external models but object using uninterpretable embeddings. Our solution is to design an interpretable embedding using DCS. These explanations are in L47 and L66. One of the advantage of our metric is its flexibility; one can proceed for the desired task and analysis. Using external models for embedding is a smart approach that removes costly manual annotation.

---

> > ### Comment · Reviewer_T7Dd · 2023-08-15
> >
> > > Presentation of paper can be improved
> >
> > Thanks to the authors for these clarifications.
> >
> > > Concept based representation.
> >
> > Thank you for taking this in account. I believe I did not make the best paper recommendation, as I do not know that literature very well, but it seems to be there exists strong links nonetheless.
> >
> > > Further exploration of attribute selection.
> >
> > Related to the previous point: if the main difference between this work and previous concept-based representations is the motivation, with this work focusing on post-hoc method for interpretation of any generative model, I believe the ability to choose any set of attributes to be one of its main advantages related to pre-existing methods. I strongly believe this should be explored.
> >
> > > More explanation for toy experiments.
> >
> > Thanks to the authors for these clarifications.
> >
> > > Text mean and Image mean / Text mean as attribute themselves.
> >
> > Than you for this additional results.
> >
> >
> > Overall, I would like to thank the authors for their answers, here and in the global rebuttal. They remove parts of my doubts and I will raise my score to 6 to reflect this.

---

### Author Rebuttal · Authors · 2023-08-10

We thank the reviewers for their valuable advice. Here, we compile reviews we want to share with all the reviewers. Please see our responses addressing the specific concerns below:



### 1. Additional experiments for providing more quantity of convincing information
#### Reviewers T7Dd, 1cWg, and ch3Y suggested adding more experiments to be more convince. We provide more experiments which support the argument of our paper.



#### 1-1. Biased image injection experiment
We conduct additional experiment with restrict desired conditions which support Sec 5.1; Correlated Image Injection Experiment. We conducted the existing experiment more densely and showed it as a graph. Please refer to **Fig. 2** in the PDF. We compare SaKLD and PaKLD with FIDs by injecting certain attributes - ("man"-"makeup" and "man"-"bangs"). Fig. 2 shows that while SaKLD and PaKLD captures distribution changes, FIDs remain about the same value. The results show that our proposed method can catch the attribute distribution change well, unlike conventional metrics.




#### 1-2. Additional model performance descriptions
We provide more results of models - LDM, StyleSwin and ProjectedGAN in the following table. These results show that our metric is similar to the tendency of FIDs.
Especially ProjectedGAN, which was criticized [FID-clip] for not focusing on fidelity, only focusing on getting good FID score by training to match training set's imagenet encoder output stats, showed inferior results in SaKLD and especially PaKLD on our metric. It indicates that even if the model can mimic few of attributes contained in imagenet labels, it is hard to catch the correlation of attributes on the training set. **Fig. 4** in the PDF shows some examples of ProjectedGAN; a baby with beard and some unusual samples.

[FID-clip] Kynkäänniemi et al., The Role of ImageNet Classes in Fréchet Inception Distance, ICLR, 2023

#### Table 1 (*The scale of SaKLD,PaKLD are 1e-7, using 50k FFHQ)
|          | StyleGAN1 | StyleGAN2 | StyleGAN3 | iDDPM | LDM   | StyleSwin | ProjectedGAN |
|:----------:|:-----------:|:-----------:|:-----------:|:-------:|:-------:|:-----------:|:--------------:|
| SaKLD    | 9.82      | 6.70      | 6.00      | 10.70 | 18.32 | 16.93     | 11.45        |
| PaKLD    | 23.50     | 16.75     | 15.50     | 25.00 | 39.71 | 40.25     | 28.18        |
| FID      | 4.74      | 3.17      | 3.20      | 7.31  | 11.86 | 5.45      | 4.45         |
| FID_CLIP | 3.17      | 1.47      | 1.66      | 2.39  | 3.57  | 3.63      | 2.45         |





### 2. Improving the way to obtain $C_{\mathcal{T}}$, the center of images in text respect.
#### Reviewer T7Dd provided constructive discussion about using the attributes themselves for obtaining $C_{\mathcal{T}}$.

Thanks to reviwer T7Dd, we improve the way to obtain $C_{\mathcal{T}}$ using the attributes themselves. We conduct the same experiment, reporting classifying accuracy by using CelebA annotation, for comparing between using captions and attribute themselves.

| | CS | DCS w/ captions | DCS w/ attribute themselves |
| :--------: |:------: | :--------: | :--------: |
| accuracy |    0.395   | 0.409    | 0.442     |

Using attributes themselves for obtaining $C_{\mathcal{T}}$ shows superior accuracy than others.
We have conducted experiments in the paper using the attributes themselves, and we confirm that the trends are all the same including Table 1 obtained with attributes themselves. Thanks for the great constructive suggestion, and we are confident that we will update all the results for the camera-ready version and that there are no changes in the aspect of tendency.

### 3. Moving interpretable analysis from appendix to Section 5.
Reviewers o9jM and E51D suggested showing a more fine-grained analysis of the scores.

Thanks to the great suggestions of reviewers, we decided to move the analysis in the appendix into the main paper. We agree that this analysis can greatly highlight the interpretable advantages of our method. The following is what has been transferred. Please refer to Fig. S13 and Tab. S7 for details.

SaKLD

SaKLD directly measures the differences in attribute distributions, indicating the challenge for models to match the density of the highest-scoring attributes to that of the training dataset. Examining the top-scoring attributes, all three StyleGAN models have similar high scores in terms of scale. However, there are slight differences, particularly in StyleGAN3, where the distribution of larger accessories such as eyeglasses or hats differs.

In contrast, iDDPM demonstrates notable scores, with attributes ‘makeup’ and ‘woman’ showing scores over two times higher than GANs. Particularly, apart from these two attributes, the remaining attributes are similar to GANs, highlighting significant differences in the density of ‘woman’ and ‘makeup’. Investigating how the generation process of diffusion models, which involve computing gradients for each pixel, affects attributes such as ‘makeup’ and ‘woman’ would be an intriguing avenue for future research.

#### PaKLD

PaKLD provides a quantitative measure of the appropriateness of relationships between attributes. Thus, if a model generates an excessive or insufficient number of specific attributes, it affects not only SaKLD but also PaKLD. Therefore, it is natural to expect that attribute pairs with high PaKLD scores will often include top-ranking attributes in SaKLD.

Nevertheless, PaKLD reveals interesting findings. Firstly, it is noteworthy that attributes related to ‘beard’ consistently receive high scores across all StyleGAN 1, 2, and 3 models. Figure S16 confirm that ‘beard’ significantly contributes to the overall PaKLD scores. This indicates that GANs generally fail to learn the relationship between beards and other attributes, making it an intriguing research topic to explore the extent of this mislearning and its underlying reasons.

---

> ### Author Response · Authors · 2023-08-11
> **Erratum**
>
> Erratum: Table1 on Author Rebuttal, the position of StyleSwin and ProjectedGAN should be changed in first row.
> Sorry for the inconvenience.
>
> Corrected version
>
> |          | StyleGAN1 | StyleGAN2 | StyleGAN3 | iDDPM | LDM   | ProjectedGAN | StyleSwin |
> |:----------:|:-----------:|:-----------:|:-----------:|:-------:|:-------:|:-----------:|:--------------:|
> | SaKLD    | 9.82      | 6.70      | 6.00      | 10.70 | 18.32 | 16.93     | 11.45        |
> | PaKLD    | 23.50     | 16.75     | 15.50     | 25.00 | 39.71 | 40.25     | 28.18        |
> | FID      | 4.74      | 3.17      | 3.20      | 7.31  | 11.86 | 5.45      | 4.45         |
> | FID_CLIP | 3.17      | 1.47      | 1.66      | 2.39  | 3.57  | 3.63      | 2.45         |

---

### Author Response · Authors · 2023-08-22
**Thanks to the Reviewers**

Dear reviewers and AC,

We sincerely thank all the reviewers for their valuable comments and suggestions, which have helped improve our paper. Although some reviewers did not engage in our response, we have incorporated all comments into both the main paper and the appendix.

We are happy that our key contributions are properly recognized:

* $\textit{Directional}$ CLIPScore allows users to grasp the presence of attributes in a given image, while CLIPScore does not.
* Single and paired attribute alignment (SaKLD and PaKLD) allow users to choose a model that captures desired attributes in the training distribution.

Sincerely,

authors

---

### Decision · Program_Chairs · 2023-09-21

**Decision:**

Reject

**Comment:**

This paper addresses an important & contemporary problem for evaluating generative models by proposing interpretable metrics (grounded in text attributes). Overall the approach is interesting and the entailed results indicate the utility of this mechanism in downstream interpretation of the models. However, the paper can benefit from more detailed analyses and improvements as indicated by the reviewers. In particular, since a large portion of the paper (and motivation) is based on grounding the model on interpretable (textual) attributes, a more detailed analysis & study of how the “choice” of attributes contributes to the model interpretability is warranted. The reviewers also raised an important point about correlation with human judgment. Since the major focus & contribution of this paper is around evaluation including human judgements would help substantiate & strengthen the approach wrt “interpretability”. The authors note (in their discussion comment) that doing this at scale is impractical. However, demonstrating this even on a smaller size (say, few thousand examples) should still be feasible and can strengthen the contribution of the paper. Another useful angle to explore would be to dive deeper into the generalizability of this approach for different tasks & model choices. The authors did provide some clarifications in their rebuttal & subsequent discussions on few other issues raised by reviewers incl. experimental results, which is helpful and appreciated.

Overall, the paper presents interesting work and a useful direction to explore; the authors are encouraged to use the feedback to further strengthen the paper for a future submission.